# Near Neighbor: Who is the Fairest of Them All?

**Sariel Har-Peled**
University of Illinois at Urbana-Champaign
Champaign, IL 61801
sariel@illinois.edu

**Sepideh Mahabadi**
Toyota Technological Institute at Chicago
Chicago, IL 60637
mahabadi@ttic.edu

## Abstract

In this work we study a fair variant of the near neighbor problem. Namely, given a set of $n$ points $P$ and a parameter $r$, the goal is to preprocess the points, such that given a query point $q$, any point in the $r$-neighborhood of the query, i.e., $\mathbb{B}(q, r)$, have the same probability of being reported as the near neighbor.

We show that $\mathsf{LSH}$ based algorithms can be made fair, without a significant loss in efficiency. Specifically, we show an algorithm that reports a point in the $r$-neighborhood of a query $q$ with almost uniform probability. The query time is proportional to $O\big(\mathrm{dns}(q.r)\mathcal{Q}(n, c)\big)$, and its space is $O(\mathcal{S}(n, c))$, where $\mathcal{Q}(n, c)$ and $\mathcal{S}(n, c)$ are the query time and space of an $\mathsf{LSH}$ algorithm for $c$-approximate near neighbor, and $\mathrm{dns}(q, r)$ is a function of the local density around $q$.

Our approach works more generally for sampling uniformly from a sub-collection of sets of a given collection and can be used in a few other applications. Finally, we run experiments to show performance of our approach on real data.

## 1   Introduction

Nowadays, many important decisions, such as college admissions, offering home loans, or estimating the likelihood of recidivism, rely on machine learning algorithms. There is a growing concern about the fairness of the algorithms and creating bias toward a specific population or feature [HPS16, Cho17, MSP16, KLL$^+$17]. While algorithms are not inherently biased, nevertheless, they may amplify the already existing biases in the data. Hence, this concern has led to the design of fair algorithms for many different applications, e.g., [DOBD$^+$18, ABD$^+$18, PRW$^+$17, CKLV19, EJJ$^+$19, OA18, CKLV17, BIO$^+$19, BCN19, KSAM19].

Bias in the data used for training machine learning algorithms is a monumental challenge in creating fair algorithms [HGB$^+$07, TE11, ZVGRG17, Cho17]. Here, we are interested in a somewhat different problem, of handling the bias introduced by the data-structures used by such algorithms. Specifically, data-structures may introduce bias in the data stored in them, and the way they answer queries, because of the way the data is stored and how it is being accessed. Such a defect leads to selection bias by the algorithms using such data-structures. It is natural to want data-structures that do not introduce a selection bias into the data when handling queries.

The target as such is to derive data-structures that are bias-neutral. To this end, imagine a data-structure that can return, as an answer to a query, an item out of a set of acceptable answers. The purpose is then to return uniformly a random item out of the set of acceptable outcomes, without explicitly computing the whole set of acceptable answers (which might be prohibitively expensive).

Several notions of fairness have been studied, including *group fairness*[1] (where demographics of the population is preserved in the outcome) and *individual fairness* (where the goal is to treat individuals

with similar conditions similarly) [DHP$^+$12]. In this work, we study the near neighbor problem from the perspective of individual fairness.

Near Neighbor is a fundamental problem that has applications in many areas such as machine learning, databases, computer vision, information retrieval, and many others, see [SDI06, AI08] for an overview. The problem is formally defined as follows. Let $(\mathcal{M}, d)$ be a metric space. Given a set $P \subseteq \mathcal{M}$ of $n$ points and a parameter $r$, the goal of the *near neighbor* problem is to preprocess $P$, such that for a query point $q \in \mathcal{M}$, one can report a point $p \in P$, such that $d(p, q) \leqslant r$ if such a point exists. As all the existing algorithms for the *exact* variant of the problem have either space or query time that depends exponentially on the ambient dimension of $\mathcal{M}$, people have considered the approximate variant of the problem. In the *c-approximate near neighbor* (ANN) problem, the algorithm is allowed to report a point $p$ whose distance to the query is at most $cr$ if a point within distance $r$ of the query exists, for some prespecified constant $c > 1$.

Perhaps the most prominent approach to get an ANN data structure is via Locality Sensitive Hashing (LSH) [IM98, HIM12], which leads to sub-linear query time and sub-quadratic space. In particular, for $\mathcal{M} = \mathbb{R}^d$, by using LSH one can get a query time of $n^{\rho+o(1)}$ and space $n^{1+\rho+o(1)}$ where for the $L_1$ distance metric $\rho = 1/c$ [IM98, HIM12], and for the $L_2$ distance metric $\rho = 1/c^2 + o_c(1)$ [AI08]. The idea of the LSH method is to hash all the points using several hash functions that are chosen randomly, with the property that closer points have a higher probability of collision than the far points. Therefore, the closer points to a query have a higher probability of falling into a bucket being probed than far points. Thus, reporting a random point from a random bucket computed for the query, produces a distribution that is biased by the distance to the query: closer points to the query have a higher probability of being chosen.

**When random nearby is better than nearest.** The bias mentioned above towards nearer points is usually a good property, but is not always desirable. Indeed, consider the following scenarios:

(I) The nearest neighbor might not be the best if the input is noisy, and the closest point might be viewed as an unrepresentative outlier. Any point in the neighborhood might be then considered to be equivalently beneficial. This is to some extent why $k$-NN classification [ELL09] is so effective in reducing the effect of noise.

(II) However, $k$-NN works better in many cases if $k$ is large, but computing the $k$ nearest-neighbors is quite expensive if $k$ is large [HAAA14]. Computing quickly a random nearby neighbor can significantly speed-up such classification.

(III) We are interested in annonymizing the query [Ada07], thus returning a random near-neighbor might serve as first line of defense in trying to make it harder to recover the query. Similarly, one might want to anonymize the nearest-neighbor [QA08], for applications were we are interested in a "typical" data item close to the query, without identifying the nearest item.

(IV) If one wants to estimate the number of items with a desired property within the neighborhood, then the easiest way to do it is via uniform random sampling from the neighborhood. In particular, this is useful for density estimation [KLK12].

(V) Another natural application is simulating a random walk in the graph where two items are connected if they are in distance at most $r$ from each other. Such random walks are used by some graph clustering algorithms [HK01].

## 1.1 Results

Our goal is to solve the near-neighbor problem, and yet be fair among "all the points" in the neighborhood. We introduce and study the ***fair near neighbor*** problem – where the goal is to report any point of $N(q, r)$ with uniform distribution. That is, report a point within distance $r$ of the query point with probability of $\mathbb{P}(q, r) = 1/n(q, r)$, where $n(q, r) = |N(q, r)|$. Naturally, we study the approximate fair near neighbor problem, where one can hope to get efficient data-structures. We have the following results:

(I) **Exact neighborhood.** We present a data structure for reporting a neighbor according to an "almost uniform" distribution with space $\mathcal{S}(n, c)$, and query time $\widetilde{O}\big(\mathcal{Q}(n, c) \cdot \frac{n(q, cr)}{n(q, r)}\big)$, where $\mathcal{S}(n, c)$ and $\mathcal{Q}(n, c)$ are, respectively, the space and query time of the standard $c$-ANN data structure. Note that, the query time of the algorithm might be high if the approximate neigh-

borhood of the query is much larger than the exact neighborhood.[2] Guarantees of this data structure hold *with high probability*. See Lemma 4.9 for the exact statement.

(II) **Approximate neighborhood.** This formulation reports an almost uniform distribution from an approximate neighborhood $S$ of the query. We can provide such a data structure that uses space $\mathcal{S}(n, c)$ and whose query time is $\widetilde{O}(\mathcal{Q}(n, c))$, albeit *in expectation*. See Lemma 4.3 for the exact statement.

Moreover, the algorithm produces the samples independently of past queries. In particular, one can assume that an adversary is producing the set of queries and has full knowledge of the data structure. Even then the generated samples have the same (almost) uniform guarantees. Furthermore, we remark that the new sampling strategy can be embedded in the existing LSH method to achieve unbiased query results. Finally, we remark that to get a distribution that is $(1 + \varepsilon)$-uniform (See preliminaries for the definition), the dependence of our algorithms on $\varepsilon$ is only $O(\log(1/\varepsilon))$.

Very recently, independent of our work, [APS19] also provides a similar definition for the fair near neighbor problem.

**Experiments.** Finally, we compare the performance of our algorithm with the algorithm that uniformly picks a bucket and reports a random point, on the MNIST, SIFT10K, and GloVe data sets. Our empirical results show that while the standard LSH algorithm fails to fairly sample a point in the neighborhood of the query, our algorithm produces an empirical distribution which is much closer to the uniform distribution: it improves the statistical distance to the uniform distribution by a significant factor.

## 2 Preliminaries

**Neighborhood, fair nearest-neighbor, and approximate neighborhood.** Let $(\mathcal{M}, \mathrm{d})$ be a metric space and let $P \subseteq \mathcal{M}$ be a set of $n$ points. Let $\mathbb{B}(c, r) = \{x \in \mathcal{M} \mid \mathrm{d}(c, x) \leqslant r\}$ be the (close) ball of radius $r$ around a point $c \in \mathcal{M}$, and let $N(c, r) = \mathbb{B}(c, r) \cap P$ be the $r$-***neighborhood*** of $c$ in $P$. The *size* of the $r$-neighborhood is $n(c, r) = |N(c, r)|$.

**Definition 2.1 (FANN).** Given a data set $P \subseteq \mathcal{M}$ of $n$ points and a parameter $r$, the goal is to preprocess $P$ such that for a given query $q$, one reports each point $p \in N(q, r)$ with probability $\mu_p$ where $\mu$ is an approximately uniform probability distribution: $\mathbb{P}(q, r)/(1+\varepsilon) \leqslant \mu_p \leqslant (1+\varepsilon)\mathbb{P}(q, r)$, where $\mathbb{P}(q, r) = 1/n(q, r)$.

**Definition 2.2 (FANN with approximate neighborhood).** Given a data set $P \subseteq \mathcal{M}$ of $n$ points and a parameter $r$, the goal is to preprocess them such that for a given query $q$, one reports each point $p \in S$ with probability $\mu_p$ where $\varphi/(1 + \varepsilon) \leqslant \mu_p \leqslant (1 + \varepsilon)\varphi$, where $S$ is a point set such that $N(q, r) \subseteq S \subseteq N(q, cr)$, and $\varphi = 1/|S|$.

**Set representation.** Let $\mathcal{U}$ be an underlying ground set of $n$ objects (i.e., elements). In this paper, we deal with sets of objects. Assume that such a set $X \subseteq \mathcal{U}$ is stored in some reasonable data-structure, where one can insert delete, or query an object in constant time. Querying for an object $o \in \mathcal{U}$, requires deciding if $o \in X$. Such a representation of a set is straightforward to implement using an array to store the objects, and a hash table. This representation allows random access to the elements in the set, or uniform sampling from the set.

If hashing is not feasible, one can just use a standard dictionary data-structure – this would slow down the operations by a logarithmic factor.

**Subset size estimation.** We need the following standard estimation tool, [BHR+17, Lemma 2.8].

**Lemma 2.3.** *Consider two sets $B \subseteq U$, where $n = |U|$. Let $\xi, \gamma \in (0, 1)$ be parameters, such that $\gamma < 1/\log n$. Assume that one is given an access to a membership oracle that, given an element $x \in U$, returns whether or not $x \in B$. Then, one can compute an estimate $s$, such that $(1 - \xi)|B| \leqslant s \leqslant (1 + \xi)|B|$, and computing this estimates requires $O((n/|B|)\xi^{-2} \log \gamma^{-1})$ oracle queries. The returned estimate is correct with probability $\geqslant 1 - \gamma$.*

**Weighted sampling.** We need the following standard data-structure for weighted sampling.

**Lemma 2.4.** *Given a set of objects $\mathcal{H} = \{o_1, \ldots, o_t\}$, with associated weights $w_1, \ldots, w_t$, one can preprocess them in $O(t)$ time, such that one can sample an object out of $\mathcal{H}$. The probability of an object $o_i$ to be sampled is $w_i / \sum_{j=1}^{t} w_j$. In addition the data-structure supports updates to the weights. An update or sample operation takes $O(\log t)$ time. (Proof in Appendix A.1)*

## 3 Data-structure: Sampling from the union of sets

**The problem.** Assume you are given a data-structure that contains a large collection $\mathcal{F}$ of sets of objects. The sets in $\mathcal{F}$ are not necessarily disjoint. The task is to preprocess the data-structure, such that given a sub-collection $\mathcal{G} \subseteq \mathcal{F}$ of the sets, one can quickly pick uniformly at random an object from the set $\bigcup \mathcal{G} := \bigcup_{X \in \mathcal{G}} X$.

**Naive solution.** The naive solution is to take the sets under consideration (in $\mathcal{G}$), compute their union, and sample directly from the union set $\bigcup \mathcal{G}$. Our purpose is to do (much) better – in particular, the goal is to get a query time that depends logarithmically on the total size of all sets in $\mathcal{G}$.

### 3.1 Preprocessing

For each set $X \in \mathcal{F}$, we build the set representation mentioned in the preliminaries section. In addition, we assume that each set is stored in a data-structure that enables easy random access or uniform sampling on this set (for example, store each set in its own array). Thus, for each set $X$, and an element, we can decide if the element is in $X$ in constant time.

### 3.2 Uniform sampling via exact degree computation

The query is a family $\mathcal{G} \subseteq \mathcal{F}$, and define $m = |\mathcal{G}| := \sum_{X \in \mathcal{G}} |X|$ (which should be distinguished from $g = |\mathcal{G}|$ and from $n = |\bigcup \mathcal{G}|$). The **degree** of an element $x \in \bigcup \mathcal{G}$, is the number of sets of $\mathcal{G}$ that contains it – that is, $\mathsf{d}_{\mathcal{G}}(x) = |\mathsf{D}_{\mathcal{G}}(x)|$, where $\mathsf{D}_{\mathcal{G}}(x) = \{X \in \mathcal{G} \mid x \in X\}$. The algorithm repeatedly does the following:

(I) Picks one set from $\mathcal{G}$ with probabilities proportional to their sizes. That is, a set $X \in \mathcal{G}$ is picked with probability $|X|/m$.
(II) It picks an element $x \in X$ uniformly at random.
(III) Computes the degree $\mathsf{d} = \mathsf{d}_{\mathcal{G}}(x)$.
(IV) Outputs $x$ and stop with probability $1/\mathsf{d}$. Otherwise, continues to the next iteration.

**Lemma 3.1.** *Let $n = |\bigcup \mathcal{G}|$ and $g = |\mathcal{G}|$. The above algorithm samples an element $x \in \bigcup \mathcal{G}$ according to the uniform distribution. The algorithm takes in expectation $O(gm/n) = O(g^2)$ time. The query time is takes $O(g^2 \log n)$ with high probability. (Proof in Appendix A.2)*

### 3.3 Almost uniform sampling via degree approximation

The bottleneck in the above algorithm is computing the degree of an element. We replace this by an approximation.

**Definition 3.2.** Given two positive real numbers $x$ and $y$, and a parameter $\varepsilon \in (0, 1)$, the numbers $x$ and $y$ are $\varepsilon$-**approximation** of each other, denoted by $x \approx_{\varepsilon} y$, if $x/(1 + \varepsilon) \leqslant y \leqslant x(1 + \varepsilon)$ and $y/(1 + \varepsilon) \leqslant x \leqslant y(1 + \varepsilon)$.

In the approximate version, given an item $x \in \bigcup \mathcal{G}$, we can approximate its degree and get an improved runtime for the algorithm.

**Lemma 3.3.** *The input is a family of sets $\mathcal{F}$ that one can preprocess in linear time. Let $\mathcal{G} \subseteq \mathcal{F}$ be a sub-family and let $n = |\bigcup \mathcal{G}|$, $g = |\mathcal{G}|$, and $\varepsilon \in (0, 1)$ be a parameter. One can sample an element $x \in \bigcup \mathcal{G}$ with almost uniform probability distribution. Specifically, the probability of an element to be output is $\approx_{\varepsilon} 1/n$. After linear time preprocessing, the query time is $O\left(g\varepsilon^{-2} \log n\right)$, in expectation, and the query succeeds with high probability. (Proof in Appendix A.3)*

**Remark 3.4.** The query time of Lemma 3.3 deteriorates to $O\left(g\varepsilon^{-2} \log^2 n\right)$ if one wants the bound to hold with high probability. This follows by restarting the query algorithm if the query time exceeds (say by a factor of two) the expected running time. A standard application of Markov's

inequality implies that this process would have to be restarted at most $O(\log n)$ times, with high probability.

**Remark 3.5.** The sampling algorithm is independent of whether or not we fully know the underlying family $\mathcal{F}$ and the sub-family $\mathcal{G}$. This means the past queries do not affect the sampled object reported for the query $\mathcal{G}$. Therefore, the almost uniform distribution property holds in the presence of several queries and independently for each of them.

### 3.4 Further Improvement.

In Appendix B, we show how to further improve the dependence on $\varepsilon$, from $\varepsilon^{-2}$ down to $\log(1/\varepsilon)$.

**Remark 3.6.** Similar to Remark 3.4, the query time of this algorithm (Lemma B.3) can be made to work with high probability with an additional logarithmic factor. Thus with high probability, the query time is $O(g\log(g/\varepsilon)\log n)$.

Finally, in Appendix C, we show further applications of this data structure.

### 3.5 Handling outliers

Imagine a situation where we have a marked set of outliers $\mathcal{O}$. We are interested in sampling from $\bigcup\mathcal{G}\backslash\mathcal{O}$. We assume that the total degree of the outliers in the query is at most $m_\mathcal{O}$ for some pre-specified parameter $m_\mathcal{O}$. More precisely, we have $\mathsf{d}_\mathcal{G}(\mathcal{O}) = \sum_{x\in\mathcal{O}}\mathsf{d}_\mathcal{G}(x) \leqslant m_\mathcal{O}$.

**Lemma 3.7.** *The input is a family of sets $\mathcal{F}$ that one can preprocess in linear time. A query is a sub-family $\mathcal{G}\subseteq\mathcal{F}$, a set of outliers $\mathcal{O}$, a parameter $m_\mathcal{O}$, and a parameter $\varepsilon\in(0,1)$. One can either*

- (A) *Sample an element $x\in\bigcup\mathcal{G}\backslash\mathcal{O}$ with $\varepsilon$-approximate uniform distribution. Specifically, the probabilities of two elements to be output is the same up to a factor of $1\pm\varepsilon$.*
- (B) *Alternatively, report that $\mathsf{d}_\mathcal{G}(\mathcal{O}) > m_\mathcal{O}$.*

*The expected query time is $O(m_\mathcal{O} + g\log(N/\varepsilon))$, and the query succeeds with high probability, where $g = |\mathcal{G}|$, and $N = |\mathcal{F}|$. (Proof in Appendix A.4)*

## 4 In the search for a fair near neighbor

In this section, we employ our data structure of Section 3 to show the two results on uniformly reporting a neighbor of a query point mentioned in Section 1.1. First, let us briefly give some preliminaries on LSH. We refer the reader to [HIM12] for further details. Throughout the section, we assume that our metric space, admits the LSH data structure.

### 4.1 Background on LSH

**Locality Sensitive Hashing (LSH).** Let $\mathcal{D}$ denote the data structure constructed by LSH, and let $c$ denote the approximation parameter of LSH. The data-structure $\mathcal{D}$ consists of $L$ hash functions $g_1, \ldots, g_L$ (e.g., $L \approx n^{1/c}$ for a $c$-approximate LSH), which are chosen via a random process and each function hashes the points to a set of buckets. For a point $p \in \mathcal{M}$, let $H_i(p)$ be the bucket that the point $p$ is hashed to using the hash function $g_i$. The following are standard guarantees provided by the LSH data structure [HIM12].

**Lemma 4.1.** *For a given query point $q$, let $S = \bigcup_i H_i(q)$. Then for any point $p \in N(q,r)$, we have that with a probability of least $1 - 1/e - 1/3$, we have (i) $p \in S$ and (ii) $|S\backslash\mathbb{B}(q,cr)| \leqslant 3L$, i.e., the number of outliers is at most $3L$. Moreover, the expected number of outliers in any single bucket $H_i(q)$ is at most $1$.*

Therefore, if we take $t = O(\log n)$ different data structures $\mathcal{D}_1, \ldots, \mathcal{D}_t$ with corresponding hash functions $g_i^j$ to denote the $i$th hash function in the $j$th data structure, we have the following lemma.

**Lemma 4.2.** *Let the query point be $q$, and let $p$ be any point in $N(q,r)$. Then, with high probability, there exists a data structure $\mathcal{D}_j$, such that $p \in S = \bigcup_i H_i^j(q)$ and $|S\backslash\mathbb{B}(q,cr)| \leqslant 3L$.*

By the above, the space used by LSH is $\mathcal{S}(n,c) = \tilde{O}(n \cdot L)$ and the query time is $\mathcal{Q}(n,c) = \tilde{O}(L)$.

## 4.2 Approximate Neighborhood

For $t = O(\log n)$, let $\mathcal{D}_1, \ldots, \mathcal{D}_t$ be data structures constructed by LSH. Let $\mathcal{F}$ be the set of all buckets in all data structures, i.e., $\mathcal{F} = \{H_i^j(p) \mid i \leqslant L, j \leqslant t, p \in P\}$. For a query point $q$, consider the family $\mathcal{G}$ of all buckets containing the query, i.e., $\mathcal{G} = \{H_i^j(q) \mid i \leqslant L, j \leqslant t\}$, and thus $|\mathcal{G}| = O(L \log n)$. Moreover, we let $\mathcal{O}$ to be the set of outliers, i.e., the points that are farther than $cr$ from $q$. Note that as mentioned in Lemma 4.1, the expected number of outliers in each bucket of LSH is at most $1$. Therefore, by Lemma 3.7, we immediately get the following result.

**Lemma 4.3.** *Given a set $P$ of $n$ points and a parameter $r$, we can preprocess it such that given query $q$, one can report a point $p \in S$ with probability $\mu_p$ where $\varphi/(1 + \varepsilon) \leqslant \mu_p \leqslant (1 + \varepsilon)\varphi$, where $S$ is a point set such that $N(q, r) \subseteq S \subseteq N(q, cr)$, and $\varphi = 1/|S|$. The algorithm uses space $\mathcal{S}(n, c)$ and its expected query time is $\widetilde{O}(\mathcal{Q}(n, c) \cdot \log(1/\varepsilon))$. (Proof in Appendix A.5)*

Remark 4.4. For the $L_1$ distance, the runtime of our algorithm is $\widetilde{O}(n^{(1/c)+o(1)})$ and for the $L_2$ distance, the runtime of our algorithm is $\widetilde{O}(n^{(1/c^2)+o(1)})$. These matches the runtime of the standard LSH-based near neighbor algorithms up to polylog factors.

## 4.3 Exact Neighborhood

As noted earlier, the result of the previous section only guarantees a query time which holds in expectation. Here, we provide an algorithm whose query time holds *with high probability*. Note that, here we cannot apply Lemma 3.7 directly, as the total number of outliers in our data structure might be large with non-negligible probability (and thus we cannot bound $m_{\mathcal{O}}$). However, as noted in Lemma 4.2, with high probability, there exists a subset of these data structures $J \subseteq [t]$ such that for each $j \in J$, the number of outliers in $S_j = \bigcup_i H_i^j(q)$ is at most $3L$, and moreover, we have that $N(q, r) \subseteq \bigcup_{j \in J} S_j$. Therefore, on a high level, we make a guess $J'$ of $J$, which we initialize it to $J' = [t]$, and start by drawing samples from $\mathcal{G}$; once we encounter more than $3L$ outliers from a certain data structure $\mathcal{D}_j$, we infer that $j \notin J$, update the value of $J' = J' \backslash \{j\}$, and set the weights of the buckets corresponding to $\mathcal{D}_j$ equal to $0$, so that they will never participate in the sampling process. As such, at any iteration of the algorithm we are effectively sampling from $\mathcal{G} = \{H_i^j(q) \mid i \leqslant L, j \in J'\}$.

**Preprocessing.** We keep $t = O(\log n)$ LSH data structures which we refer to as $\mathcal{D}_1, \ldots, \mathcal{D}_t$, and we keep the hashed points by the $i$th hash function of the $j$th data structure in the array denoted by $H_i^j$. Moreover, for each bucket in $H_i^j$, we store its size $|H_i^j|$.

**Query Processing.** We maintain the variables $z_i^j$ showing the weights of the bucket $H_i^j(q)$, which is initialized to $|H_i^j(q)|$ that is stored in the preprocessing stage. Moreover, we keep the set of outliers detected from $H_i^j(q)$ in $\mathcal{O}_i^j$ which is initially set to be empty. While running the algorithm, as we detect an outlier in $H_i^j(q)$, we add it to $\mathcal{O}_i^j$, and we further decrease $z_i^j$ by one. Moreover, in order to keep track of $J'$, for any data structure $\mathcal{D}_j$, whenever $\sum_i |\mathcal{O}_i^j|$ exceeds $3L$, we will ignore all buckets in $\mathcal{D}_j$, by setting all corresponding $z_i^j$ to zero.

At each iteration, the algorithm proceeds by sampling a bucket $H_i^j(q)$ proportional to its weight $z_i^j$, but only among the set of buckets from those data structures $\mathcal{D}_j$ for which less than $3L$ outliers are detected so far, i.e., $j \in J'$. We then sample a point uniformly at random from the points in the chosen bucket that have not been detected as an outlier, i.e., $H_i^j(q) \backslash \mathcal{O}_i^j$. If the sampled point is an outlier, we update our data structure accordingly. Otherwise, we proceed as in Lemma B.3.

Definition 4.5 (Active data structures and active buckets). Consider an iteration $k$ of the algorithm. Let us define the set of *active data structures* to be the data structures from whom we have seen less than $3L$ outliers so far, and let us denote their indices by $J_k' \subseteq [t]$, i.e., $J_k' = \{j \mid \sum_i |\mathcal{O}_j^i| < 3L\}$.

Moreover, let us define the *active buckets* to be all buckets containing the query in these active data structures, i.e., $\mathcal{G}_k = \{H_i^j(q) \mid i \leqslant L, j \in J_k'\}$.

**Observation 4.6.** *Lemma 4.2 implies that with high probability at any iteration $k$ of the algorithm $N(q, r) \subseteq \bigcup \mathcal{G}_k$.*

**Definition 4.7** (active size). For an active bucket $H_i^j(q)$, we define its active size to be $z_i^j$ which shows the total number of points in the bucket that have not yet been detected as an outlier, i.e., $|H_i^j(q) \backslash \mathcal{O}_i^j|$.

**Lemma 4.8.** *Given a set $P$ of $n$ points and a parameter $r$, we can preprocess it such that given a query $q$, one can report a point $p \in P$ with probability $\mu_p$, so that there exists a value $\rho \in [0,1]$ where*

- *For $p \in N(q, r)$, we have $\frac{\rho}{(1+O(\varepsilon))} \leqslant \mu_p \leqslant (1 + O(\varepsilon))\rho$.*
- *For $p \in N(q, cr) \backslash N(q, r)$, we have $\mu_p \leqslant (1 + O(\varepsilon))\rho$.*
- *For $p \notin N(q, cr)$, we have $\mu_p = 0$.*

*The space used is $\widetilde{O}(\mathcal{S}(n,c))$ and the query time is $\widetilde{O}\big(\mathcal{Q}(n,c) \cdot \log(1/\varepsilon)\big)$ with high probability. (Proof in Appendix A.6)*

**Lemma 4.9.** *Given a set $P$ of $n$ points and a parameter $r$, we can preprocess it such that given a query $q$, one can report a point $p \in S$ with probability $\mu_p$ where $\mu$ is an approximately uniform probability distribution: $\varphi/(1 + \varepsilon) \leqslant \mu_p \leqslant \varphi(1 + \varepsilon)$, where $\varphi = 1/|N(q,r)|$. The algorithm uses space $\mathcal{S}(n, c)$ and has query time of $\widetilde{O}\big(\mathcal{Q}(n,c) \cdot \frac{|N(q,cr)|}{|N(q,r)|} \cdot \log(1/\varepsilon)\big)$ with high probability. (Proof in Appendix A.7)*

# 5   Experiments

In this section, we consider the task of retrieving a random point from the neighborhood of a given query point, and evaluate the effectiveness of our proposed algorithm empirically on real data sets.

**Data set and Queries.** We run our experiments on three datasets that are standard benchmarks in the context of Nearest Neighbor algorithms (see [ABF17])

(I) Our first data set contains a random subset of 10K points in the MNIST training data set [LBBH98][3]. The full data set contains 60K images of hand-written digits, where each image is of size 28 by 28. For the query, we use a random subset of $100$ (out of 10K) images of the MNIST test data set. Therefore, each of our points lie in a $784$ dimensional Euclidean space and each coordinate is in $[0, 255]$.

(II) Second, we take SIFT10K image descriptors that contains 10K 128-dimensional points as data set and 100 points as queries [4].

(III) Finally, we take a random subset of 10K words from the GloVe data set [PSM14] and a random subset of 100 words as our query. GloVe is a data set of 1.2M word embeddings in 100-dimensional space and we further normalize them to unit norm.

We use the $L_2$ Euclidean distance to measure the distance between the points.

**LSH data structure and parameters.** We use the locality sensitive hashing data structure for the $L_2$ Euclidean distance [AI08]. That is, each of the $L$ hash functions $g_i$, is a concatenation of $k$ unit hash functions $h_i^1 \oplus \cdots \oplus h_i^k$. Each of the unit hash functions $h_i^j$ is chosen by selecting a point in a random direction (by choosing every coordinate from a Gaussian distribution with parameters $(0, 1)$). Then all the points are projected onto this one dimensional direction. Then we put a randomly shifted one dimensional grid of length $w$ along this direction. The cells of this grid are considered as buckets of the unit hash function. For tuning the parameters of LSH, we follow the method described in [DIIM04], and the manual of E2LSH library [And05], as follows.

For MNIST, the average distance of a query to its nearest neighbor in the our data set is around $4.5$. Thus we choose the near neighbor radius $r = 5$. Consequently, as we observe, the $r$-neighborhood of at least half of the queries are non-empty. As suggested in [DIIM04] to set the value of $w = 4$, we tune it between 3 and 5 and set its value to $w = 3.1$. We tune $k$ and $L$ so that the false negative rate (the near points that are not retrieved by LSH) is less than $10\%$, and moreover the cost of hashing (proportional to $L$) balances out the cost of scanning. We thus get $k = 15$ and $L = 100$. This also agrees with the fact that $L$ should be roughly square root of the total number of points. Note that we use a single LSH data structure as opposed to taking $t = O(\log n)$ instances. We use the same

method for the other two data sets. For SIFT, we use $R = 255$, $w = 4$, $k = 15$, $L = 100$, and for GloVe we use $R = 0.9$, $w = 3.3$, $k = 15$, and $L = 100$.

**Algorithms.** Given a query point $q$, we retrieve all $L$ buckets corresponding to the query. We then implement the following algorithms and compare their performance in returning a neighbor of the query point.

- **Uniform/Uniform**: Picks bucket uniformly at random and picks a random point in bucket.
- **Weighted/Uniform**: Picks bucket according to its size, and picks uniformly random point inside bucket.
- **Optimal**: Picks bucket according to size, and then picks uniformly random point $p$ inside bucket. Then it computes $p$'s degree *exactly* and rejects $p$ with probability $1 - 1/deg(p)$.
- **Degree approximation**: Picks bucket according to size, and picks uniformly random point $p$ inside bucket. It approximates $p$'s degree and rejects $p$ with probability $1 - 1/deg'(p)$.

**Degree approximation method.** We use the algorithm of Appendix B for the degree approximation: we implement a variant of the sampling algorithm which repeatedly samples a bucket uniformly at random and checks whether $p$ belongs to the bucket. If the first time this happens is at iteration $i$, then it outputs the estimate as $deg'(p) = L/i$.

**Experiment Setup.** In order to compare the performance of different algorithms, for each query $q$, we compute $M(q)$: the set of neighbors of $q$ which fall to the same bucket as $q$ by at least one of the $L$ hash functions. Then for $100|M(q)|$ times, we draw a sample from the neighborhood of the query, using all four algorithms. We compare the empirical distribution of the reported points on $|M(q)|$ with the uniform distribution on it. More specifically, we compute the total variation distance (statistical distance)[5] to the uniform distribution. We repeat each experiment 10 times and report the average result of all 10 experiments over all 100 query points.

**Results.** Figure 1 shows the comparison between all four algorithms. To compare their performance, we compute the total variation distance of the empirical distribution of the algorithms to the uniform distribution. For the tuned parameters ($k = 15$ , $L = 100$), our results are as follows. For MNIST, we see that our proposed degree approximation based algorithm performs only 2.4 times worse than the optimal algorithm, while we see that other standard sampling methods perform 6.6 times and 10 times worse than the optimal algorithm. For SIFT, our algorithm performs only 1.4 times worse than the optimal while the other two perform 6.1 and 9.7 times worse. For GloVe, our algorithm performs only 2.7 times worse while the other two perform 6.5 and 13.1 times worse than the optimal algorithm.

Moreover, in order get a different range of degrees and show that our algorithm works well for those cases, we further vary the parameters $k$ and $L$ of LSH. More precisely, to get higher ranges of the degrees, first we decrease $k$ (the number of unit hash functions used in each of the $L$ hash function); this will result in more collisions. Second, we increase $L$ (the total number of hash functions). These are two ways to increase the degree of points. For example for the MNIST data set, the above procedure increases the degree range from $[1, 33]$ to $[1, 99]$.

**Query time discussion.** As stated in the experiment setup, in order to have a meaningful comparison between distributions, in our code, we retrieve a random neighbor of each query $100m$ times, where m is the size of its neighborhood (which itself can be as large as 1000). We further repeat each experiment 10 times. Thus, every query might be asked upto $10^6$ times. This is going to be costly for the optimal algorithm that computes the degree exactly. Thus, we use the fact that we are asking the same query many times and preprocess the exact degrees for the optimal solution. Therefore, it is not meaningful to compare runtimes directly. Thus we run the experiments on a smaller size dataset to compare the runtimes of all the four approaches: For $k = 15$ and $L = 100$, our sampling approach is twice faster than the optimal algorithm, and almost five times slower than the other two approaches. However, when the number of buckets (L) increases from 100 to 300, our algorithm is 4.3 times faster than the optimal algorithm, and almost 15 times slower than the other two approaches.

**Trade-off of time and accuracy.** We can show a trade-off between our proposed sampling approach and the optimal. For the MNIST data set with tuned parameters ($k = 15$ and $L = 100$), by asking twice more queries (for degree approximation), the solution of our approach improves from 2.4 to 1.6, and with three times more, it improves to 1.2, and with four times more, it improves to 1.05. For

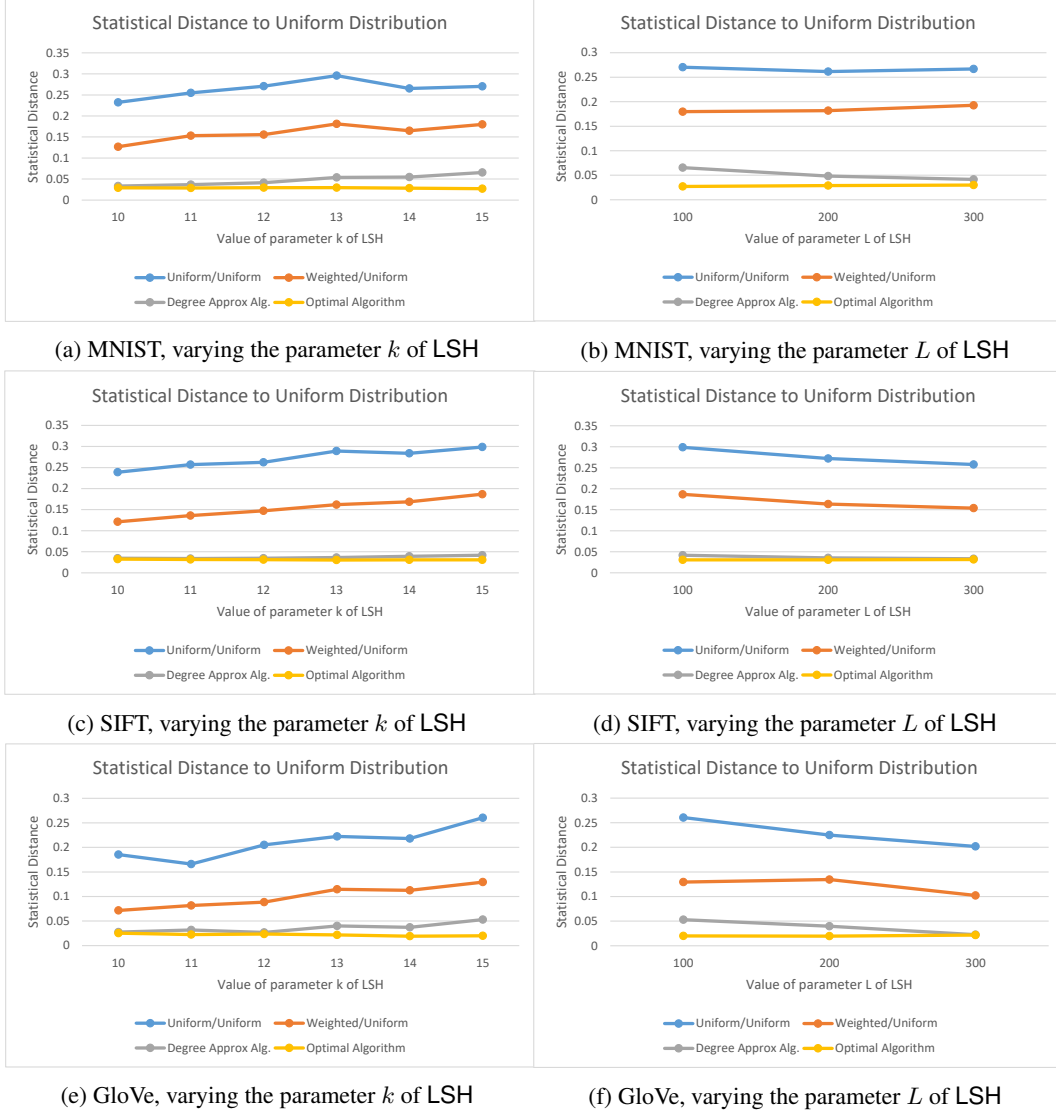

(a) MNIST, varying the parameter $k$ of LSH

(b) MNIST, varying the parameter $L$ of LSH

(c) SIFT, varying the parameter $k$ of LSH

(d) SIFT, varying the parameter $L$ of LSH

(e) GloVe, varying the parameter $k$ of LSH

(f) GloVe, varying the parameter $L$ of LSH

Figure 1: Comparison of the performance of the four algorithms is measured by computing the statistical distance of their empirical distribution to the uniform distribution.

the SIFT data set (using the same parameters), using twice more queries, the solution improves from 1.4 to 1.16, and with three times more, it improves to 1.04, and with four times more, it improves to 1.05. For GloVe, using twice more queries, the solution improves from 2.7 to 1.47, and with three times more, it improves to 1.14, and with four times more, it improves to 1.01.

# 6   Acknowledgement

The authors would like to thank Piotr Indyk for the helpful discussions about the modeling and experimental sections of the paper.

## Footnotes

[1]The concept is denoted as statistical fairness too, e.g., [Cho17].

[2]As we show, the term $\mathcal{Q}(n, r) \cdot \frac{n(q, cr)}{n(q, r)}$ can also be replaced by $\mathcal{Q}(n, r) + |N(q, cr) \backslash N(q, r)|$ which can potentially be smaller.

[3]The dataset is available here: http://yann.lecun.com/exdb/mnist/

[4]The dataset if available here: http://corpus-texmex.irisa.fr/

[5]For two discrete distributions $\mu$ and $\nu$ on a finite set $X$, the total variation distance is $\frac{1}{2} \sum_{x \in X} |\mu(x) - \nu(x)|$.

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
