[Supplementary Material]

# Near Neighbor: Who is the Fairest of Them All?
# (Appendix)

**Sariel Har-Peled**
University of Illinois at Urbana-Champaign
Champaign, IL 61801
sariel@illinois.edu

**Sepideh Mahabadi**
Toyota Technological Institute at Chicago
Chicago, IL 60637
mahabadi@ttic.edu

## A   Omitted Proofs

### A.1   Proof of Lemma 2.4

*Proof:* Build a balanced binary tree $T$, where the objects of $\mathcal{G}$ are stored in the leaves. Every internal node $u$ of $T$, also maintains the total weight $w(u)$ of the objects in its subtree. The tree $T$ has height $O(\log t)$, and weight updates can be carried out in $O(\log t)$ time, by updating the path from the root to the leaf storing the relevant object.

Sampling is now done as follows – we start the traversal from the root. At each stage, when being at node $u$, the algorithm considers the two children $u_1, u_2$. It continues to $u_1$ with probability $w(u_1)/w(u)$, and otherwise it continues into $u_2$. The object sampled is the one in the leaf that this traversal ends up at. ∎

### A.2   Proof of Lemma 3.1

*Proof:* Let $m = |\mathcal{G}|$. Observe that an element $x \in \bigcup \mathcal{G}$ is picked by step (II) with probability $\alpha = \mathsf{d}(x)/m$. The element $x$ is output with probability $\beta = 1/\mathsf{d}(x)$. As such, the probability of $x$ to be output by the algorithm in this round is $\alpha\beta = 1/|\mathcal{G}|$. This implies that the output distribution is uniform on all the elements of $\bigcup \mathcal{G}$.

The probability of success in a round is $n/m$, which implies that in expectation $m/n$ rounds are used, and with high probability $O((m/n) \log n)$ rounds. Computing the degree $\mathsf{d}_{\mathcal{G}}(x)$ takes $O(|\mathcal{G}|)$ time, which implies the first bound on the running time. As for the second bound, observe that an element can appear only once in each set of $\mathcal{G}$, which readily implies that $\mathsf{d}(y) \leqslant |\mathcal{G}|$, for all $y \in \bigcup \mathcal{G}$. ∎

### A.3   Proof of Lemma 3.3

*Proof:* Let $m = |\mathcal{G}|$. Since $\mathsf{d}(x) = |\mathsf{D}_{\mathcal{G}}(x)|$, it follows that we need to approximate the size of $\mathsf{D}_{\mathcal{G}}(x)$ in $\mathcal{G}$. Given a set $X \in \mathcal{G}$, we can in constant time check if $x \in X$, and as such decide if $X \in \mathsf{D}_{\mathcal{G}}(x)$. It follows that we can apply the algorithm of Lemma 2.3, which requires $W(x) = O\left(\frac{g}{\mathsf{d}(x)}\varepsilon^{-2}\log n\right)$ time, where the algorithm succeeds with high probability. The query algorithm is the same as before, except that it uses the estimated degree.

For $x \in \bigcup \mathcal{G}$, let $\mathcal{E}_x$ be the event that the element $x$ is picked for estimation in a round, and let $\mathcal{E}'_x$ be the event that it was actually output in that round. Clearly, we have $\mathbb{P}[\mathcal{E}'_x \mid \mathcal{E}_x] = 1/d$, where $d$ is the degree estimate of $x$. Since $d \approx_\varepsilon \mathsf{d}(x)$ (with high probability), it follows that $\mathbb{P}[\mathcal{E}'_x \mid \mathcal{E}_x] \approx_\varepsilon 1/\mathsf{d}(x)$. Since there are $\mathsf{d}(x)$ copies of $x$ in $\mathcal{G}$, and the element for estimation is picked uniformly from the sets of $\mathcal{G}$, it follows that the probability of any element $x \in \bigcup \mathcal{G}$ to be

output in a round is

$$\mathbb{P}\big[\mathcal{E}'_x\big] = \mathbb{P}\big[\mathcal{E}'_x \mid \mathcal{E}_x\big]\,\mathbb{P}\big[\mathcal{E}_x\big] = \mathbb{P}\big[\mathcal{E}'_x \mid \mathcal{E}_x\big]\,\frac{\mathsf{d}(x)}{m}\ \approx_\varepsilon\ 1/m,$$

as $\mathcal{E}'_x \subseteq \mathcal{E}_x$. As such, the probability of the algorithm terminating in a round is $\alpha = \sum_{x \in \bigcup \mathcal{G}} \mathbb{P}[\mathcal{E}'_x] \approx_\varepsilon n/m \geqslant n/2m$. As for the expected amount of work in each round, observe that it is proportional to

$$W = \sum_{x \in \bigcup \mathcal{G}} \mathbb{P}[\mathcal{E}_x] W(x) = \sum_{x \in \bigcup \mathcal{G}} \frac{\mathsf{d}(x)}{m} \frac{g}{\varepsilon^2 \mathsf{d}(x)} \log n = O\Big(\frac{ng}{m}\varepsilon^{-2} \log n\Big).$$

Intuitively, since the expected amount of work in each iteration is $W$, and the expected number of rounds is $1/\alpha$, the expected running time is $O(W/\alpha)$. This argument is not quite right, as the amount of work in each round effects the probability of the algorithm to terminate in the round (i.e., the two variables are not independent). We continue with a bit more care – let $L_i$ be the running time in the $i$th round of the algorithm if it was to do an $i$th iteration (i.e., think about a version of the algorithm that skips the experiment in the end of the iteration to decide whether it is going to stop), and let $Y_i$ be a random variable that is 1 if the (original) algorithm had not stopped at the end of the first $i$ iterations of the algorithm.

By the above, we have that $y_i = \mathbb{P}[Y_i = 1] = \mathbb{P}[Y_i = 1 \mid Y_{i-1} = 1]\,\mathbb{P}[Y_{i-1} = 1] \leqslant (1 - \alpha)y_{i-1} \leqslant (1 - \alpha)^i$, and $\mathbb{E}[L_i] = O(W)$. Importantly, $L_i$ and $Y_{i-1}$ are independent (while $L_i$ and $Y_i$ are dependent). We clearly have that the running time of the algorithm is $O\big(\sum_{i=1}^\infty Y_{i-1}L_i\big)$ (here, we define $Y_0 = 1$). Thus, the expected running time of the algorithm is proportional to

$$\mathbb{E}\Big[\sum_i Y_{i-1}L_i\Big] = \sum_i \mathbb{E}[Y_{i-1}L_i] = \sum_i \mathbb{E}[Y_{i-1}]\,\mathbb{E}[L_i] \leqslant W \sum_i y_{i-1} \leqslant W \sum_{i=1}^\infty (1 - \alpha)^{i-1} = \frac{W}{\alpha}$$
$$= O(g\varepsilon^{-2} \log n),$$

because of linearity of expectations, and since $L_i$ and $Y_{i-1}$ are independent. ∎

## A.4 Proof of Lemma 3.7

*Proof:* The main modification of the algorithm is that whenever we encounter an outlier (the assumption is that one can check if an element is an outlier in constant time), then we delete it from the set $X$ where it was discovered. If we implement sets as arrays, this can be done by moving an outlier object to the end of the active prefix of the array, and decreasing the count of the active array. We also need to decrease the (active) size of the set. If the algorithm encounters more than $m_\mathcal{O}$ outliers then it stops and reports that the number of outliers is too large.

Otherwise, the algorithm continues as before. The only difference is that once the query process is done, the active count (i.e., size) of each set needs to be restored to its original size, as is the size of the set. This clearly can be done in time proportional to the query time. ∎

## A.5 Proof of Lemma 4.3

*Proof:* Let $S = \bigcup \mathcal{G} \setminus \mathcal{O}$; by Lemma 4.2, we know that $N(q, r) \subseteq S \subseteq N(q, cr)$, and moreover in expectation $m_\mathcal{O} \leqslant L = |\mathcal{G}|$. We apply the algorithm of Lemma 3.7. The runtime of the algorithm is in expectation $\widetilde{O}(|\mathcal{G}| \log(1/\varepsilon)) = \widetilde{O}(L \cdot \log(1/\varepsilon)) = \widetilde{O}(\mathcal{Q}(n, c) \cdot \log(1/\varepsilon))$, and the algorithm produces an almost uniform distribution over the points in $S$. ∎

## A.6 Proof of Lemma 4.8

*Proof:* First note that the algorithm never outputs an outlier, and thus the third item is always satisfied. Next, let $K$ be a random variable showing the number of iterations of the algorithm, and for an iteration $k$, define the random variable $M_k = N(q, cr) \cap \bigcup \mathcal{G}_k$ as the set of non-outlier points in the set of active buckets. Conditioned on $K = k$, by Lemma B.3, we know that the distribution of the output is almost uniform on $M_k$. Moreover, we know that for all $k$ we have $M_k \subseteq M_{k-1}$, and that by Observation 4.6, $N(q, r) \subseteq M_k$. Therefore, for all points in $N(q, r)$ their probability of being reported as the final output of the algorithm is equal, and moreover, for all points in $N(q, cr) \setminus N(q, r)$,

their probability of being reported is lower (as at some iteration, some of these points might go out of the set of active buckets). This proves the probability condition.

To bound the query time, let us consider the iterations where the sampled point $p$ is an outlier, and not an outlier, separately. The total number of iterations where an outlier point is sampled is at most $3L \cdot t = \widetilde{O}(L) = \widetilde{O}(\mathcal{Q}(n,c))$ for which we only pay $\widetilde{O}(1)$ cost. For non-outlier points, their total cost can be bounded using Lemma B.3 (and Remark 3.6) by $\widetilde{O}(|\mathcal{G}_1| \cdot \log(g/\varepsilon)) = \widetilde{O}(L \cdot \log(1/\varepsilon)) = \widetilde{O}(\mathcal{Q}(n,c) \cdot \log(1/\varepsilon))$. ∎

### A.7 Proof of Lemma 4.9

*Proof:* We run Algorithm of Lemma 4.8, and while its output is outside of $N(q, r)$, we ignore it and run the algorithm again. By Lemma 4.8, the output is guaranteed to be almost uniform on $N(q, r)$. Moreover, by Lemma 4.8, and because with high probability, we only need to run the algorithm $\widetilde{O}(\frac{|N(q,cr)|}{|N(q,r)|})$ times, we get the desired bound on the query time. ∎

## B  Almost uniform sampling via simulation

Here, we show how one can avoid the degree approximation stage in the algorithm of Section 3.3, and achieve only a polylogarithmic dependence on $\varepsilon^{-1}$. To this end, let $x$ be the element picked. We need to simulate a process that accepts $x$ with probability $1/\mathsf{d}(x)$.

We start with the following natural idea for estimating $\mathsf{d}(x)$ – probe the sets randomly (with replacement), and stop in the $i$th iteration if it is the first iteration where the probe found a set that contains $x$. If there are $g$ sets, then the distribution of $i$ is geometric, with probability $p = \mathsf{d}(x)/g$. In particular, in expectation, $\mathbb{E}[i] = g/\mathsf{d}(x)$, which implies that $\mathsf{d}(x) = g/\mathbb{E}[i]$. As such, it is natural to take $g/i$ as an estimation for the degree of $x$. As such, since we want to simulate a process that succeeds with probability $1/\mathsf{d}(x)$, it would be natural to return 1 with probability $i/g$ and 0 otherwise. Surprisingly, while this seems like a heuristic, it does work, under the right interpretation as testified by the following.

**Lemma B.1.** *Assume we have $g$ urns, and exactly $d > 0$ of them, are non-empty. We can check if a specific urn is empty in constant time. There is a randomized algorithm, that outputs a number $Y \geq 0$, such that $\mathbb{E}[Y] = 1/d$. The expected running time of the algorithm is $O(g/d)$.*

*Proof:* The algorithm repeatedly probes urns (uniformly at random), until it finds a non-empty urn. Assume it found a non-empty urn in the $i$th probe. The algorithm outputs the value $i/g$ and stops.

Setting $p = d/g$, and let $Y$ be the output of the algorithm. we have that

$$\mathbb{E}[Y] = \sum_{i=1}^{\infty} \frac{i}{g}(1-p)^{i-1}p = \frac{p}{g(1-p)}\sum_{i=1}^{\infty} i(1-p)^i = \frac{p}{g(1-p)} \cdot \frac{1-p}{p^2} = \frac{1}{pg} = \frac{1}{d},$$

using the formula $\sum_{i=1}^{\infty} ix^i = \frac{x}{(1-x)^2}$.

The expected number of probes performed by the algorithm until it finds a non-empty urn is $1/p = g/d$, which implies that the expected running time of the algorithm is $O(g/d)$. ∎

The natural way to deploy Lemma B.1, is to run its algorithm to get a number $y$, and then return 1 with probability $y$. The problem with this idea is that $y$ can be strictly larger than 1, which is meaningless for probabilities. Instead, we backoff by using the value $y/\Delta$, for some parameter $\Delta$. If the returned value is larger than 1, we just treat it at zero. If the zeroing never happened, the algorithm would return one with probability $1/(\mathsf{d}(x)\Delta)$ – which we can use to our purposes via, essentially, amplification. Instead, the probability of success is going to be slightly smaller, but fortunately, the loss can be made arbitrarily small by taking $\Delta$ to be sufficiently large.

**Lemma B.2.** *There are $g$ urns, and exactly $d > 0$ of them are not empty. Furthermore, assume one can check if a specific urn is empty in constant time. Let $\gamma \in (0,1)$ be a parameter. There is a randomized algorithm, that outputs a number $Z \geq 0$, such that $Z \in [0,1]$, and $\mathbb{E}[Z] \in I = \left[\frac{1}{d\Delta} - \gamma, \frac{1}{d\Delta}\right]$, where $\Delta = \lceil \ln \gamma^{-1} \rceil + 4 = \Theta(\log \gamma^{-1})$. The expected running time of the algorithm is $O(g/d)$.*

*Alternatively, the algorithm can output a bit $X$, such that $\mathbb{P}[X = 1] \in I$.*

*Proof:* We modify the algorithm of Lemma B.1, so that it outputs $i/(g\Delta)$ instead of $i/g$. If the algorithm does not stop in the first $g\Delta + 1$ iterations, then the algorithm stops and outputs $0$. Observe that the probability that the algorithm fails to stop in the first $g\Delta$ iterations, for $p = d/g$, is $(1 - p)^{g\Delta} \leqslant \exp\left(-\frac{d}{g}g\Delta\right) \leqslant \exp(-d\Delta) \leqslant \exp(-\Delta) \ll \gamma$.

Let $Z$ be the random variable that is the number output by the algorithm. Arguing as in Lemma B.1, we have that $\mathbb{E}[Z] \leqslant 1/(d\Delta)$. More precisely, we have $\mathbb{E}[Z] = \frac{1}{d\Delta} - \sum_{i=g\Delta+1}^{\infty} \frac{i}{g\Delta}(1-p)^{i-1}p$. Let

$$\sum_{i=gj+1}^{g(j+1)} \frac{i}{g}(1-p)^{i-1}p \leqslant (j+1)\sum_{i=gj+1}^{g(j+1)}(1-p)^{i-1}p = (j+1)(1-p)^{gj}\sum_{i=0}^{g-1}(1-p)^{i}p$$

$$\leqslant (j+1)(1-p)^{gj} \leqslant (j+1)\left(1 - \frac{d}{g}\right)^{gj} \leqslant (j+1)\exp(-dj)$$

Let $g(j) = \frac{j+1}{\Delta}\exp(-dj)$. We have that $\mathbb{E}[Z] \geqslant \frac{1}{d\Delta} - \beta$, where $\beta = \sum_{j=\Delta}^{\infty} g(j)$. Furthermore, for $j \geqslant \Delta$, we have

$$\frac{g(j+1)}{g(j)} = \frac{(j+2)\exp(-d(j+1))}{(j+1)\exp(-dj)} \leqslant \left(1 + \frac{1}{\Delta}\right)e^{-d} \leqslant \frac{5}{4}e^{-d} \leqslant \frac{1}{2}.$$

As such, we have that

$$\beta = \sum_{j=\Delta}^{\infty} g(j) \leqslant 2g(\Delta) \leqslant 2\frac{\Delta+1}{\Delta}\exp(-d\Delta) \leqslant 4\exp(-\Delta) \leqslant \gamma,$$

by the value of $\Delta$. This implies that $\mathbb{E}[Z] \geqslant 1/(d\Delta) - \beta \geqslant 1/(d\Delta) - \gamma$, as desired.

The alternative algorithm takes the output $Z$, and returns $1$ with probability $Z$, and zero otherwise. ∎

**Lemma B.3.** *The input is a family of sets $\mathcal{F}$ that one preprocesses in linear time. Let $\mathcal{G} \subseteq \mathcal{F}$ be a sub-family and let $n = |\bigcup\mathcal{G}|$, $g = |\mathcal{G}|$, and let $\varepsilon \in (0,1)$ be a parameter. One can sample an element $x \in \bigcup\mathcal{G}$ with almost uniform probability distribution. Specifically, the probability of an element to be output is $\approx_{\varepsilon} 1/n$. After linear time preprocessing, the query time is $O(g\log(g/\varepsilon))$, in expectation, and the query succeeds, with high probability (in $g$).*

*Proof:* The algorithm repeatedly samples an element $x$ using steps (I) and (II) of the algorithm of Section 3.2. The algorithm returns $x$ if the algorithm of Lemma B.2, invoked with $\gamma = (\varepsilon/g)^{O(1)}$ returns $1$. We have that $\Delta = \Theta(\log(g/\varepsilon))$. Let $\alpha = 1/(\mathsf{d}(x)\Delta)$. The algorithm returns $x$ in this iteration with probability $p$, that is in the range $[\alpha - \gamma, \alpha]$. Observe that $\alpha > 1/(g\Delta)$, which implies that $\gamma \ll (\varepsilon/4)\alpha$, it follows that $p \approx_{\varepsilon} 1/(d\Delta)$, as desired. The expected running time of each round is $O(g/\mathsf{d}(x))$.

Arguing as in Lemma 3.3., this implies that each round, in expectation takes $O(ng/m)$ time, where $m = |\mathcal{G}|$. Similarly, the expected number of rounds, in expectation, is $O(\Delta m/n)$. Again, arguing as in Lemma 3.3, implies that the expected running time is $O(g\Delta) = O(g\log(g/\varepsilon))$. ∎

## C Applications

Here are a few examples of applications of a data-structure for sampling from a sub-collection of sets:

(A) Given a subset $X$ of vertices in the graph, randomly pick (with uniform distribution) a neighbor to one of the vertices of $X$. This can be used in simulating disease spread [KE05].

(B) In this paper, we use a variant of this data-structure to implement the fair ANN.

(C) Uniform sampling for range searching [HQT14, AW17, AP19]. Indeed, consider a set of points, stored in a data-structure for range queries. Using the above, we can support sampling from the points reported by several queries, even if the reported answers are not disjoint.

Being unaware of any previous work on this problem, we believe this data-structure is of independent interest.