[Reviews · NeurIPS 2019]

Reviewer 1



The writing up to page 4 is very wordy and I believe could be more succinctly written. First of all, I am not entirely sure why the problem of sampling from sub collection of sets need to be repeated twice in the paper. In addition, the paper should clearly state that the new sampling strategy can be embedded in the existing LSH method to achieve unbiased query results. Nevertheless, the algorithm does seem interesting - the key bottleneck of estimating the degree of a particular point (basically the number of distinct buckets that contain it) is identified and there are interesting solutions based on existing work in this paper. Section 3 - line 161 states the union of $G$ but $G$ is a set of sets. I can see that the part after the equal sign does make sense. I see that there are similar notational mistakes throughout the paper. Section 4 - line 226: what is $x$ and its relationship to $p$? Section 5 - Please clarify the following experiment setting: "Our data set contains the first 10000 points in the MNIST training data set..." Does choosing the first X points in the dataset help reproducibility? I am not sure why randomization was not used. Also should comment the distribution of the classes in the dataset that was used in the experiment. - Comparison is largely in terms of quality of the query results. Could the authors comment on the querying time and time-accuracy tradeoff? Originality - The sampling framework seems novel. Quality - The paper overall has average quality and somewhat incomplete experimental results. Clarity - The paper could be re-written to deliver the message more succinctly because some definitions are repeated throughout the paper. Significance - The problem/framework the paper discusses is of practical importance but because the paper is more empirically oriented (although the authors may disagree and do provide proof sketches in the appendix), it should provide more compete experimental results.

Reviewer 2



This paper considers a "fair" variant of the nearest-neighbor problem: given a set of points D, one aims to build a data structure so that, given a query point q, and a radius r, the data structure can be used to return an (almost) uniform at random point of the intersection between D and the ball of radius r centered on q. Such a data structure could be used to avoid "unfair" applications of nearest neighbor approaches (e.g., applications were the same point was to be returned at every query). The authors present a reduction showing that, if one has a LSH that returns a c-approximate NN in time T, using space S, then one can turn that algorithm into one that returns an almost UAR point in the (approximate) ball of the chosen radius with space O(S) and expected time O(T). To prove the reduction the authors introduce a new problem that they solve efficiently: a collection C of sets is given at the outset, and given a subcollection S \subseteq C, one has to return a UAR element from the union of the sets in S. The authors show that one can answer such a query in time O(|C|), by rejection sampling (together with an efficient algorithm for approximating an element degree). The authors then apply the solution to this problem to the fair NN problem, obtaining the aforementioned result (together with some variants). Finally, the authors test their algorithm on a number of datasets. I think this is a very neat paper, and I recommend acceptance.

Reviewer 3



In this paper the authors provide a new formulation for fair nearest neighbors. The key idea is that instead of reporting the nearest neighbor, or an approximate nearest neighbor, one reports uniformly at random a point within the ball of radius r. The authors provide a data structure of independent interest that essentially samples according to the l0 norm (#distinct elements) in the union of a query. This primitive is used in combination with an LSH black box to design the desired data structures. Then, the proposed method is evaluated on a real-world dataset. From a technical point of view, section 3 and the corresponding formulation are the main technical contribution in my opinion, as well as the fair NN formulation the main originality of the paper. Overall, it is also well written. [I thank the authors for the detailed response. I have upgraded my score from 5 to 6]

[Author Response · NeurIPS 2019]

We thank all the reviewers for their constructive feedback and comments. We will include the new experiments in full scale for the final version of the paper, and will incorporate all comments and suggestions on the clarity of presentation.

**To reviewer #1:** Our experiments were focused on varying the number and size of the buckets (by changing the parameters $k$ and $L$). Per your suggestion, we ran our experiments for two other datasets: SIFT10K and GloVe (random subset of 10k words out of 1.2M words). These along with MNIST are standard benchmarks in the context of Nearest Neighbor algorithms (see [ABF17]). We use the same tuning method for both datasets and further vary $k$ and $L$ to get different size buckets. The end results are slightly better than the results on the MNIST. For example, for the same tuned parameter ($k = 15$ and $L = 100$), we get the following. For SIFT, our algorithm performs 1.35 times worse than the optimal algorithm (with the same measure of statistical distance to the uniform distribution), while the other two approaches perform 6.04 times worse and 9.62 times worse than the optimal. For GloVe, our algorithm performs 2.42 times worse but the other two perform 5.94 and 11.84 times worse respectively.

We used the first 10K vectors in MNIST in order to reduce the amount of randomization. The number of occurances of the digits from 0 to 9 are $[1001, 1127, 991, 1032, 980, 863, 1014, 1070, 944, 978]$. We further ran the experiments again with the random selection of the entries and here are the results for the base case of $k = 15$ and $L = 100$: Our approach performs 2.42 times worse, while the other two approaches perform 6.52 times and 9.87 times worse than the optimal.

We will revise the introduction part of the paper to make it more succint and postpone the discussion about importance and other applications of sampling from sub-collection of sets to the appendix.

In order to have a meaningful comparison between distributions, in our code, we retrieve a random neighbor of each query $100m$ times, where m is the size of its neighborhood (which itself can be as large as 1000). We further repeat each experiment 10 times. Thus, every query might be asked upto $10^6$ times. This is going to be costly for the optimal algorithm that computes the degree exactly. Thus, we use the fact that we are asking the same query many times and preprocess the exact degrees for the optimal solution. Therefore, it was not meaningful to compare runtimes directly. We are running the experiments on a smaller size dataset just to show the difference in the runtimes of all the four approaches: Our sampling approach is twice faster than the optimal algorithm, and almost five times slower than the other two approaches. However, when the number of buckets (L) increases from 100 to 300, our algorithm is 4.3 times faster than the optimal algorithm, and almost 15 times slower than the other two approaches.

As you suggest, we can show a trade-off between our proposed sampling approach and the optimal. For example, by asking twice more queries (for degree approximation), the solution of our approach improves from 2.5 to 1.58, and with three times more, it improves to 1.21, and with four times more, it improves to 1.05. We will include the full graph.

**To reviewer #4:** The main criteria of our data structure is that even if you ask the same query multiple times, you get a point which is independently almost uniform each time. In order to achieve this using $L_0$ sampler, we need to have an $L_0$ sampler per query. We dont consider a linear dependence on the number of queries to be desired as one can get a much simpler data structure (simple variant of LSH) for a linear dependence. We remark that in the $L_0$ sampler setting, the algorithm needs to work in a more restricted model.

For tuning the parameters of LSH, we follow the method described in [DIIM04] (see the references of the main submission file), and manual of E2LSH library [And05]. Nevertheless, we further vary $k$ and $L$ to get buckets of larger size, more outliers, and larger number of buckets.

Certainly one can get a tradeoff by approximating the degree more precisely. Please see above (lines 27-29).

Our sampling approach provides an algorithm with dependency on the number of points ($n$) which matches that of the optimal LSH, while providing an almost uniform distribution. However, it is an interesting open question to find the tight dependency on the density of the neighborhood of the query ($|N(q, cr)|/|N(q, r)|$).

In general, to meaningfully compare the distributions, we might ask for a random neighbor of each query $10^6$ times (please see lines 18-20 above for details). This is why we did not afford to run it on huge datasets. However, we expect that the algorithm performs similarly on larger datasets. Per your suggestion, we ran our experiments for slightly larger dataset (100k word representaion from the GloVe dataset that has smaller dimension), and the results are as follows. For the base case ($k = 15$ and $L = 100$), our approach performs 1.41 times worse than the optimal algorithm but the other two approaches perform 3.73 and 6.01 times worse.

# References

[ABF17] M. Aumüller, E. Bernhardsson, and A. Faithfull. Ann-benchmarks: A benchmarking tool for approximate nearest neighbor algorithms. In *International Conference on Similarity Search and Applications*, 2017.

[And05] Alexandr Andoni. E2lsh 0.1 user manual. *https://www.mit.edu/ andoni/LSH/manual.pdf*, 2005.


[Meta-Review · NeurIPS 2019]

Computing "fair" nearest neighbors is an interesting research question, and the authors give a simple and practical algorithm for this problem. The reviewers thank the authors for the rebuttal and would like the authors to incorporate the richer empirical results into the final version of the paper.